# Efficacy, Safety and Acceptability of a Very-Low-Energy Diet in Adolescents with Obesity: A Fast Track to Health Sub-Study

**DOI:** 10.3390/nu16183125

**Published:** 2024-09-16

**Authors:** Megan L. Gow, Hiba Jebeile, Eve T. House, Shirley Alexander, Louise A. Baur, Justin Brown, Clare E. Collins, Chris T. Cowell, Kaitlin Day, Sarah P. Garnett, Alicia Grunseit, Mary-Kate Inkster, Cathy Kwok, Sarah Lang, Susan J. Paxton, Helen Truby, Krista A. Varady, Natalie B. Lister

**Affiliations:** 1Sydney Medical School, The University of Sydney, Westmead, NSW 2145, Australia; 2Institute of Endocrinology and Diabetes, The Children’s Hospital at Westmead, Westmead, NSW 2145, Australia; 3Weight Management Services, The Children’s Hospital at Westmead, Westmead, NSW 2145, Australia; 4Department of Paediatric Endocrinology and Diabetes, Monash Children’s Hospital, Clayton, VIC 3168, Australia; 5Department of Paediatrics, Monash University, Clayton, VIC 3168, Australia; 6School of Health Sciences, College of Health, Medicine and Wellbeing, University of Newcastle, Callaghan, NSW 2308, Australia; 7Food and Nutrition Research Program, Hunter Medical Research Institute, New Lambton Heights, NSW 2305, Australia; 8Kids Research, The Children’s Hospital at Westmead, Westmead, NSW 2145, Australia; 9School of Agriculture, Food and Ecosystem Sciences, University of Melbourne, Melbourne, VIC 3010, Australia; 10Department of Nutrition, Dietetics & Food, Monash University, Melbourne, VIC 3004, Australia; 11Nutrition and Dietetics, The Children’s Hospital at Westmead, Westmead, NSW 2145, Australia; 12School of Psychology and Public Health, La Trobe University, Melbourne, VIC 3086, Australia; 13School of Primary and Allied Health Care, Monash University, Melbourne, VIC 3004, Australia; 14School of Human Movement and Nutrition Sciences, University of Queensland, Brisbane, QLD 4072, Australia; 15Department of Kinesiology and Nutrition, University of Illinois, Chicago, IL 60612, USA

**Keywords:** adolescent, obesity, dietary intervention, very-low-energy diet, safety, acceptability

## Abstract

The aim of this study was to determine the efficacy, safety and acceptability of a 4-week very-low-energy diet (VLED) program for adolescents with obesity. Adolescents (13–17 years) with obesity and ≥1 obesity-related complication were Fast Track to Health 52-week randomized controlled trial participants. Adolescents undertook a 4-week micronutrient-complete VLED (800 kcal/day), with weekly dietitian support. Anthropometric data were recorded at baseline and week-4 and side-effects at day 3–4, week-1, -2, -3 and -4. Adolescents completed an acceptability survey at week-4. A total of 134 adolescents (14.9 ± 1.2 years, 50% male) had a 5.5 ± 2.9 kg (*p* < 0.001) mean weight loss at week-4: 95% experienced ≥1 and 70% experienced ≥3 side-effects during the VLED program, especially during the first week. Hunger, fatigue, headache, irritability, loose stools, constipation and nausea were most common. Reporting more side-effects at day 3–4 correlated with greater weight loss at week-4 (r = −0.188, *p* = 0.03). Adolescents reported ‘losing weight’ (34%) and ‘prescriptive structure’ (28%) as the most positive aspects of VLED, while ‘restrictive nature’ (45%) and ‘meal replacement taste’ (20%) were least liked. A dietitian-monitored short-term VLED can be implemented safely and is acceptable for many adolescents seeking weight loss, despite frequent side-effects. Investigating predictors of acceptability and effectiveness could determine adolescents most suited to VLED programs.

## 1. Introduction

Child and adolescent obesity continues to be a global health concern, with 206 million 5–19 year olds worldwide anticipated to be affected by obesity by 2025, and 254 million by 2030 [1]. Obesity in young people is complicated by a range of associated health problems including psychosocial distress, insulin resistance, type 2 diabetes, orthopedic disorders, hypertension, dyslipidemia and metabolic-associated fatty liver disease [2,3,4,5,6], highlighting the need for effective treatments for those seeking weight management. 

In a review of clinical practice guidelines, weight loss was identified as a primary goal of treatment of pediatric obesity in the presence of obesity-related comorbidities [7]. Guidelines typically recommend that weight loss be achieved using a lifestyle modification program that incorporates nutrition and physical activity education, encouraging sustained behavior change [7,8]. Systematic reviews of pediatric obesity treatment demonstrate that conventional lifestyle modification programs lead to modest weight loss in the short-to-medium term [9,10]. However, many young people do not respond to conventional treatment [11], suggesting the need for alternate therapies for this group.

A very-low-energy diet (VLED) is an alternate therapy. VLEDs typically aim for <800 kcal/3350 kJ and <50 g carbohydrate per day, often achieved using meal replacements such as shakes and bars, which are nutritionally complete for adults. Overall, meta-analyses of both adult and child and adolescent studies indicate that VLEDs are effective for achieving rapid weight loss [12,13]. Our 2019 meta-analysis of 20 studies found VLED programs led to a mean weight loss of 10.1 kg in children and adolescents with obesity following a 3-to-20-week intervention, with 5.3 kg of weight loss maintained at follow-up 5 to 14.5 months from baseline in a meta-analysis of seven studies [12]. VLEDs may also improve cardiometabolic variables in young people, including remission of type 2 diabetes [12,14,15,16].

Despite their efficacy, concerns exist regarding the safety of VLED use in adolescents, including potential side-effects, impacts on growth, cardiac function, psycho-behavioral health and acceptability [17]. In our systematic review, only 10 of 24 studies reported monitoring for adverse effects, of which six reported their occurrence, and four reported no adverse effects [12]. Given the limited reporting of VLED side-effects to date, conclusions on their safety cannot be drawn from the existing literature [12], preventing their consistent inclusion as part of pediatric obesity clinical practice treatment guidelines [7]. Further studies with more rigorous monitoring of adverse events are required to verify the safety of using VLED programs in youth as a weight management tool. The aim of this study was to explore the efficacy, safety and acceptability of a 4-week VLED for adolescents with obesity and co-morbidities seeking weight loss treatment.

## 2. Materials and Methods

Adolescents were participants in the Fast Track to Health two-arm multi-site randomized controlled trial conducted in pediatric hospitals in Sydney and Melbourne, Australia. This study was approved by The Sydney Children’s Hospitals Network Human Research Ethics Committee (HREC/17/SCHN/164) on the 22 August 2017. Written informed consent was obtained from parents/carers and assent from adolescents. The trial was prospectively registered with the Australian New Zealand Clinical Trial Register (ACTRN12617001630303). The Fast Track to Health study protocol [18] and primary outcomes [4,19] have been published.

In brief, eligible participants were aged 13–17 years with obesity as defined by the International Obesity Task Force criteria [20] and ≥1 obesity-related complication [18]. Adolescents with conditions or undergoing treatments known to impact weight were excluded, as were non-English speaking adolescents/parents [18].

Eligible and consenting adolescents were randomized to one of two dietary interventions: intermittent energy restriction or continuous energy restriction. Participants then completed the three phases of the study: the ‘jumpstart’ VLED intervention (weeks 0–4); intensive dietary intervention (weeks 5–16) and continued intervention and/or maintenance (weeks 17–52). The primary outcome for the Fast Track to Health study was change in BMI z-score at 52-weeks by intervention group. For the secondary analyses, combined data collected from baseline to week-4 are reported for all participants who completed Phase 1 (i.e., not by intervention group). Detailed methods for Phase 2 and Phase 3 can be found in the protocol and primary outcome papers [4,18,19]. 

Rolling recruitment occurred between 2018 and 2022 with one COVID-19 related pause in recruitment during 2020 to avoid lockdown-related challenges in adhering to a VLED intervention. Major Australian holiday periods (i.e., Easter and Christmas) and individually specific religious observances were also avoided (e.g., Ramadan) for commencement of the study due to preconceived difficulties in adhering to a VLED intervention during these periods.

All participants completed Phase 1 before commencing their randomized diet intervention. A micronutrient complete VLED product (Optifast^®^ VLCD^TM^, Nestlé Health Science, Nestlé Australia Ltd., Rhodes, NSW, Australia) was provided at no cost to participants. Participants chose to either consume: four Optifast^®^ formulated meal replacement products per day (shakes, soups, bars, and/or desserts), with low-carbohydrate-content vegetables and 1 teaspoon of vegetable oil, or three Optifast^®^ formulated meal replacements and one meal consisting of 100−150 g lean, cooked meat, any amount of low-carbohydrate vegetables and 1 teaspoon of oil. These combinations provided ∼800 kcal/3350 kJ per day, with <40% of energy derived from carbohydrate (∼50 g/d), 40–55% of energy as protein, and <20% of energy as fat. The reduced carbohydrate prescription during the VLED was designed to induce ketosis. Participants were also permitted to eat ad libitum from a list of ‘low-energy’ vegetables and selected fruits and encouraged to consume at least 2 L of water or other low-energy beverages daily. Participants were encouraged to maintain usual levels of physical activity during this phase of the study. Considering this dietary prescription, rate of weight loss during this phase was anticipated to be 1–2 kg per week.

Any participant who did not tolerate the Optifast^®^ meal replacements was given the option to follow a dietitian-prescribed, food-based VLED in conjunction with two daily multivitamins, or were provided with recommendations for alternate suitable meal replacement products.

During Phase 1, participants and a parent/guardian had weekly one-on-one contact with the study dietitian who delivered the VLED intervention and reviewed dietary intake, eating behaviors and family support with dietary change. Physical activity habits were also discussed, with participants generally recommended to maintain current exercise habits during Phase 1 of the study. Contact was face-to-face at baseline, week-1 and -4. At week-2 and -3, face-to-face visits were encouraged. Appointments could be conducted remotely (phone or video call) as per participant preference or COVID-19-related restrictions, at the discretion of the dietitian. A support phone call was conducted at day 3–4, which included recording any side-effects experienced by participants. In addition to the support provided by the dietitian, participants received planned weekly standardized text messages to their mobile device to offer further advice and support for diet, physical activity or sleep.

### 2.1. Data Collection

#### 2.1.1. Efficacy

Efficacy of the intervention was assessed by examining change in weight status during the intervention. Weight and height were measured using standard free-standing procedures. BMI, BMI z-scores and BMI expressed as a percentage of the 95th percentile (BMIp95) were calculated [21,22]. Waist circumference, defined as the horizontal distance around the umbilicus using the left hand under technique, was measured to the nearest 0.1 cm using a flexible steel tape. The average of three measurements was used for data analyses. Waist-to-height ratio was calculated as a measure of central adiposity and metabolic risk [23].

Bioelectrical impedance analysis (BIA) was used to assess body composition at baseline and week-4 using a multi-frequency stand-on body composition analyzer (Sydney site: Tanita MC780MA, Tanita Corporation, Tokyo, Japan; Melbourne site: Seca mBCA515 Seca^®^, Hamburg, Germany). Participants stood with bare feet on the electrode panel and held electrodes in both hands, with the electrodes in contact with thumb and palm during the measurements [24]. 

Ketosis was monitored by finger prick analysis in a non-fasting state at the weekly dietitian visits (week-1, -2, -3 and -4) using the Freestyle Optium Neo Blood Glucose and Ketone Monitoring System (Abbott, Ireland). Ketosis was defined as a plasma concentration of β-hydroxybutyrate (βHB) ≥ 0.3 mmol/L.

#### 2.1.2. Safety

During dietetic reviews and/or support at day 3–4 and week-1, -2, -3 and -4, participants were asked whether they experienced (yes or no) the following side-effects, all of which have been previously reported in adults [25]: hunger, fatigue, headache, irritability, loose stools, constipation, nausea, lack of concentration, sensitivity to cold, bad breath, menstrual disturbances, muscle cramps, hair loss or other (not specified, but noted separately if considered an adverse event). Adverse events were recorded separately, indicating the safety of the VLED intervention.

#### 2.1.3. Acceptability

Parent and adolescent views relating to VLED acceptability were collected at week-4 using online surveys. Adolescents were asked what they liked most and least about following the VLED intervention (open-ended), whether it was difficult to go out with friends and whether family life was more difficult (yes, no, sometimes) and the acceptability (very useful, useful, neutral, not useful or not at all useful) of intervention components (dietitian reviews, support, written materials and weekly text messages). Both adolescents and parents were asked to rate, on a 100-point Likert scale, whether they found the intervention easy and enjoyable, with 0 being least easy/enjoyable and 100 being most easy/enjoyable. Parents were also asked to answer ‘All of the time; Most of the time; Some of the time; A little of the time; None of the time’ in response to a number of statements concerning the VLED intervention as related to their child (e.g., they had enough food on the meal plan; they kept to the food prescribed; they ate extra food not prescribed; they didn’t follow the meal plan; they felt good about the meal plan, etc.).

### 2.2. Data Analysis

Statistical analysis was performed using IBM SPSS Statistics, version 26.0 (Chicago, IL, USA). Data were assessed for normality and descriptive statistics were conducted to summarize adolescent anthropometric outcomes at baseline and 4-weeks, side-effects throughout the 4-week VLED, and adolescent and parent VLED acceptability (answers to open-ended acceptability questions were classified into common themes/responses) as mean ± SD for parametric data, median [range] for non-parametric data and number (%) for categorical data. Paired sample *t*-tests (parametric distribution) were performed to estimate changes over time in normally distributed outcomes. Assumptions of statistical tests were assessed and met. In all analyses, a *p*-value < 0.05 was considered statistically significant. Only those participants who completed the 4-week VLED intervention are included in this analysis to represent the full extent of expected side effects over a 4-week period. Those who withdrew from the study during the 4-week VLED are described, including in reference to experienced side-effects.

## 3. Results

The mean age of the 134 participants who completed the 4-week VLED intervention was 14.9 ± 1.2 years (50% male). Seven participants withdrew from the study before completion of the 4-week VLED phase (see primary paper [15]). Of these, three withdrew as they specifically did not wish to continue with the VLED, one of which received an additional support session at week-1 due to VLED side-effects (ongoing headaches and feeling faint) but withdrew from the study during week-3.

### 3.1. Efficacy

#### Changes in Anthropometry

Baseline and week-4 anthropometric data are reported in Table 1. Overall, weight (mean difference ± SD: -5.5 ± 2.9 kg, *p* < 0.001), BMI z-score, BMIp95, waist circumference, waist-to-height ratio and body fat percentage all significantly reduced from baseline to week-4. Individual weight change of participants is displayed in a waterfall plot (Figure 1). Most participants experienced weight loss (128 of 134, 96%) during the VLED program, ranging from 0.2 to 14.1 kg of weight loss. Six participants (4%) gained weight (mean 5.4% weight loss, range 2.8% gain to 12.4% loss). Most participants experienced a reduction in BMIp95 (130 of 133, 98%), ranging from a 1.1 to 16.9 reduction in percentage points (Figure 1).

Median [range] βHB levels were 0.5 [0.0–3.7] mmol/L at week 1 (*n* = 79), 0.5 [0.0–4.7] mmol/L at week 2 (*n* = 63), 0.4 [0.0–3.1] mmol/L at week 3 (*n* = 57) and 0.2 [0.0–3.2] mmol/L at week 4 (*n* = 102). At week-1, -2, -3 and -4, 56 of 79 (71%), 45 of 63 (71%), 39 of 57 (68%) and 47 of 102 (46%) met the threshold for ketosis (≥0.3 mmol/L βHB), respectively. In total, 109 participants had their ketone levels assessed at any time point during the VLED program, 80 of whom (73%) met the threshold for ketosis at some time point achieving 6.3 ± 2.7 kg weight loss, compared with 3.3 kg loss in those who did not achieve the threshold (t = 5.214, *p* < 0.001). Participants who were in ketosis at every visit (*n* = 20) achieved 7.35 ± 2.13 kg weight loss compared to 5.05 ± 2.94 kg weight loss in those who were not. At all time points, meeting the threshold for ketosis was associated with significantly more weight loss at week-4 (week-1: t = 2.645, *p* = 0.010; week-2: t = 4.323, *p* < 0.001; week-3: t = 2.324, *p* = 0.024; week-4: t = 5.817, *p* < 0.001).

### 3.2. Safety

#### 3.2.1. Side-Effects

There were a range of known side-effects experienced by participants throughout the 4-week VLED. The number of participants experiencing ≥ 1 side-effect (*n* = 100, 76% of 129 participants with data) and the total number of side-effects experienced peaked at week-1 with approximately double the number of side-effects reported compared with week-2, -3 and -4, and triple the number compared with day 3–4 (Figure 2). At day 3–4 and week-2, -3 and -4, 39%, 50%, 57% and 43% of participants with side-effects data experienced ≥1 side-effect. The most frequently reported side-effects were hunger, fatigue, headache, irritability, loose stools, constipation, nausea and lack of concentration (Table 2).

Only seven participants (5%) did not experience any side-effects during the 4-week VLED program. Most (70%) experienced ≥3 side-effects throughout the 4-week VLED. In some cases, this was the same side-effect throughout the 4-week VLED, and in other cases the side-effect experienced changed throughout the 4-week VLED.

#### 3.2.2. Adverse Events

In addition to VLED-related side-effects, there were 11 adverse events reported during the 4-week VLED including 7 participants who developed a viral infection (COVID-19 infection documented in two). Three participants reported dizziness, one with an episode of fainting during the VLED phase, and one participant diagnosed with an acute issue (pilonidal sinus) unrelated to the VLED intervention, for which treatment was sought. Two participants had to modify their VLED regimen due to the adverse event: one discontinued the VLED while unwell with a viral illness causing severe diarrhea, and the other discontinued VLED meal replacements while on antibiotics to treat an infection. There were no adverse events related to psychological distress or eating behaviors during the VLED phase.

### 3.3. Acceptability

#### 3.3.1. Adolescent Acceptability

When asked what they liked most about this phase of the program, 41 of 122 participants reported weight loss. Another 34 reported that the prescriptive structure of the VLED intervention made it easy to follow. Other reasons for liking the VLED intervention included the meal replacements themselves (*n* = 11), being able to eat food as well as meal replacements (*n* = 8), decreased hunger (*n* = 5), feeling better/having more energy (*n* = 5), trying new foods (*n* = 3) and having more independence/control over what they ate (*n* = 2). Two participants reported that finishing the VLED phase was the best thing about it, and another six reported liking nothing about following the VLED.

Of the 123 participants who answered what they liked least about the VLED intervention, 55 reported the restrictive nature of the VLED and lack of diet variety. Participants disliked the taste of the meal replacements products (*n* = 24), reported boredom with limited food choices (*n* = 7), difficulty sticking to the plan for four weeks (*n* = 7), difficulty going out (*n* = 5), affecting mood/energy levels (*n* = 4), hunger (*n* = 4), tiredness (*n* = 2), difficult at the beginning (*n* = 2) and cravings (*n* = 1). Ten participants reported that there was nothing that they liked least about the VLED intervention.

Fifty-eight percent of participants reported that it was difficult to go out with friends while following the VLED. Common reasons for this included not being able to eat what their friends were eating, difficulty determining what they could eat while out, not wanting their friends to know they were on a weight loss program, and managing cravings. Participants who did not have any difficulty going out with friends (42%) reported being committed to the weight loss program, knowing what options they could eat while out, eating before they went out and having supportive friends. Several participants also reported that they were not able to go out with friends while following the VLED due to COVID-19 lockdowns.

Only 6% reported that family life was ‘more difficult’ while on the VLED plan, with another 22% reporting that it was ‘sometimes more difficult’. Reasons related to not being able to eat what their siblings ate, being more irritable during the VLED phase, and not having the necessary support from their family.

Overall, 87% of participants reported that meetings with the dietitian were useful/very useful. Support phone calls and text messages from the dietitian were also well received (65% useful/very useful), as were dietitian provided written materials (e.g., meal plans and fact sheets; 86% useful/very useful). General weekly text messages sent to participants had mixed acceptability, with 47% reporting they were useful/very useful, 33% feeling neutral about them and 20% reporting that they were not useful/not at all useful.

When asked to rate on a 100-point Likert scale as to whether the VLED intervention was easy (100-points) or difficult (0-points) to follow, and enjoyable (100-points) or not enjoyable (0-points) to follow, the mean ± SD score for each was 61.1 ± 23.6 and 52.7 ± 24.3, respectively.

#### 3.3.2. Parent/Carer-Reported Acceptability

When parents/carers were asked to rate on a 100-point Likert scale whether the VLED intervention was easy (100-points) or difficult (0-points) and enjoyable (100-points) or not enjoyable (0-points) for their adolescent to follow, the mean ± SD scores were 75.1 ± 22.7 (*n* = 106) and 60.7 ± 24.1 (*n* = 96) respectively. Acceptability of specific aspects of the VLED intervention, as reported by a parent/carer, are outlined in Table 3.

## 4. Discussion

We found that a dietitian-monitored VLED program can be implemented safely in adolescents with obesity. While expected VLED-related side-effects were commonly experienced by participants, only one adverse event (a single episode of fainting) was deemed to be potentially related to the intervention. Acceptability of the VLED program was mixed. Our study addresses a current gap in the literature surrounding the safety and acceptability of a VLED intervention for use in adolescents with severe obesity and related comorbidities.

The mean weight loss of participants included in this study was 5.5 kg at the completion of the 4-week VLED phase. In comparison, the mean weight loss in our 2019 meta-analysis of 20 VLED studies in children and adolescents was 10.1 kg [12]. This greater weight loss is explained by the longer duration of most VLED interventions included in the meta-analysis: only six studies included in the meta-analysis implemented a VLED intervention of ≤4-weeks [26,27,28,29,30,31], with 11 studies included in the meta-analysis implementing a VLED intervention of ≥8-weeks [14,32,33,34,35,36,37,38,39,40,41]. Furthermore, six studies included in the meta-analysis incorporated an inpatient phase of ≥1-week [26,28,29,31,34,42], enhancing adherence to the intervention. Meta-regression and moderator analysis indicated that longer trials and trials incorporating an inpatient phase predicted greater weight loss. This study indicates that adolescents completing a short-term VLED intervention in an outpatient setting, i.e., while also attending school and participating in family life as normal, can achieve significant weight loss consistent with the anticipated rate of 1–2 kg/week. Furthermore, 73% of participants achieved ketosis at some point during the intervention, suggesting that adolescents can adhere to a VLED without the need for an inpatient stay, so long as ongoing dietitian support is provided throughout the intervention. However, only 15% of participants were in ketosis for the duration of the intervention, suggesting that further weight loss could be achieved by enhancing adherence. Importantly, this finding also demonstrates that adolescents can have meaningful weight loss without being in a constant state of ketosis, which may impact diet acceptability.

The Fast Track to Health study used a short-term VLED as a ‘jumpstart’ weight loss initiative at the beginning of a longer-term weight loss intervention trial. The rationale for this was related to an earlier study conducted by our team which found that early weight loss was a predictor of longer-term weight loss in adolescents [43], which has since been reported in several other pediatric and adult studies [11,44,45]. A 2020 study found that weight loss in the range of 2.4% to 3.4% at week-4 of a pediatric obesity treatment program predicted 10% BMI z-score reduction at 6-month follow-up [45]. In line with this previous finding, our 5.5 kg weight loss is equivalent to 5.5% of body weight (baseline weight 100 kg) and was associated with a 12.5% reduction in BMI z-score at 52-weeks as reported in the primary outcome paper [19]. This reflects maintenance of weight lost during the 4-week VLED phase of the intervention out to the end of the trial supporting the implementation of an intensive maintenance phase, as in Fast Track to Health [18], to prevent weight re-gain.

Consistent with the adult literature [25,46], our findings highlight that adolescents who participate in a VLED intervention for weight loss can expect to experience a range of known side-effects, with 95% experiencing ≥1 side-effect during the 4-week intervention. With the frequency of side-effects peaking during week-1 of the intervention, patients should be reassured that side-effects typically reduce or subside with continued adherence and appropriate hydration. Only two participants discontinued the VLED for a period during the 4-week phase; however, in both instances these participants suffered viral illnesses and so the discontinuation was clinically assessed as unrelated to the VLED intervention. Notably, no psycho-behavioural adverse events were reported. In fact, our paper focused on these outcomes found that psycho-behavioural outcomes improved overall in participants from baseline to 4-weeks [4].

Reported for the first time in the scientific literature, our acceptability surveys highlight a range of adolescent perspectives. Weight loss and the structure of the intervention were rated as the best things about the VLED program by a third of participants each. Together with 81% of parents reporting that their child felt more in control of their eating ‘all’ or ‘most’ of the time during the VLED intervention, these findings support previous research reporting that adolescents like a structured intervention for obesity treatment [47].

In contrast, there were several challenges in adhering to a VLED program highlighted by participants. These included eating out with friends, the restrictive nature of the VLED intervention and the taste of the meal replacement products. Common strategies reported to alleviate some of these challenges included having the support of family and friends and planning meals in advance. Our study unveils some of the common barriers and facilitators to adhering to a VLED intervention that can assist clinicians in adequately preparing an adolescent for some of the challenges experienced when undertaking a VLED intervention. For instance, more participants reported that scheduled one-on-one dietitian sessions (face-to-face or online) were very useful/useful compared to support calls/text messages and general text messages, suggesting the importance of regular scheduled dietitian contact to engage and support adolescents during a VLED intervention.

Interestingly, parents rated the VLED intervention as both more enjoyable, and easier, compared with ratings reported by the adolescent. This indicates the internal struggles likely to be experienced by an individual completing a VLED intervention, of which those around them may not be aware. It also highlights the need for various avenues of support, including health professionals, family and friends, to problem solve as challenges arise, and enhance overall adherence. Future research should examine predictors of both VLED effectiveness and acceptability to determine whether certain adolescents may be more suited to undertaking a VLED intervention.

It is possible that a VLED intervention could be used intermittently for rapid weight loss to assist adolescents with obesity-related complications and prevent the progression of comorbidities. For example, in the prevention of type 2 diabetes, research has demonstrated that rapid weight loss achieved following adherence to a VLED is specifically associated with reduced liver fat, a key driver of type 2 diabetes [14]. Future studies in this cohort will further elucidate how VLED-induced weight loss is associated with aspects of cardiometabolic health beyond the VLED intervention period.

A 2020 review of clinical practice guidelines found only 7 of 28 guidelines recommended intensive dietary approaches be used in the treatment of pediatric obesity in the presence of comorbidities or severe obesity [7]. In contrast, 16 guidelines made recommendations for the use of pharmacotherapy or surgical interventions when traditional lifestyle intervention was unsuccessful. Given the efficacy of a short-term health professional supervised VLED to achieve weight loss in adolescents, a VLED could be used as an intermediate step to the treatment of severe obesity or obesity-related comorbidities in adolescents, i.e., if traditional behavioral lifestyle intervention is not sufficient, before pharmacological or surgical intervention or if pharmacological/surgical interventions are not accessible. It is likely that further research is required to determine how VLED-related side-effects and acceptability track beyond a 4-week period before clinical practice guidelines can reflect our preliminary findings.

Strengths of the current study include the large sample of adolescents completing a VLED intervention and the planned and methodical collection of VLED side-effects at five time points. Limitations include the lack of a control group given that all Fast Track to Health participants completed the VLED intervention before commencing the randomized diet. Participants were not fasting when anthropometric data were collected at 4-weeks, as at baseline, suggesting that actual weight loss during the 4-week intervention may have been greater. Lack of biochemistry results at 4-weeks also meant we were unable to make timepoint-matched correlations between VLED-induced weight loss and changes in cardiometabolic blood markers. This could be considered in future research.

## 5. Conclusions

The current study highlights that a dietitian-monitored VLED can be implemented safely in the short-term and can be effective for achieving rapid short-term weight loss in adolescents with obesity. Dietitians and other health professionals should be aware of the frequency of VLED-related side-effects and mixed acceptability which may impact on individual suitability and adherence to the intervention without adequate support. 

## Figures and Tables

**Figure 1 nutrients-16-03125-f001:**
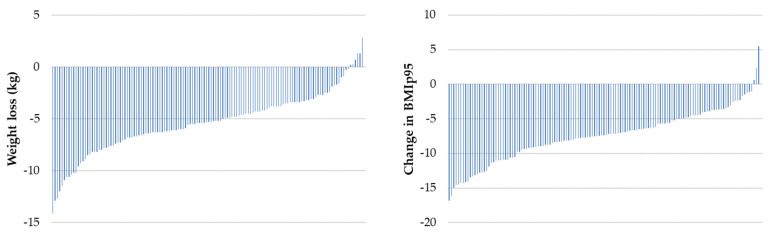
Individual participant changes in weight (kg) and BMI expressed as a percentage of the 95th percentile (BMIp95) during the 4-week very-low-energy diet phase.

**Figure 2 nutrients-16-03125-f002:**
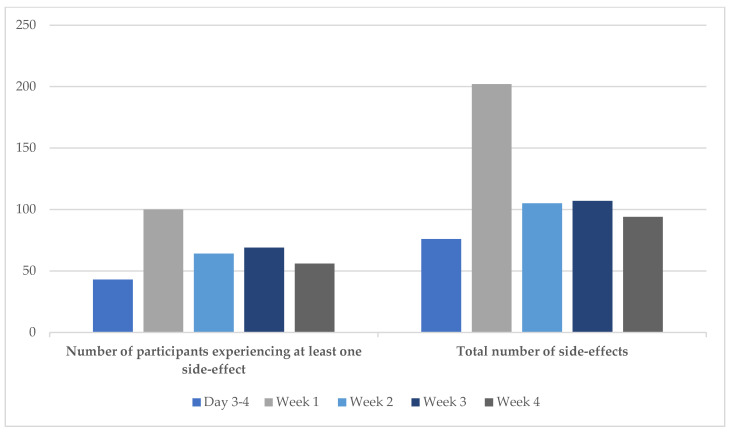
Number of participants experiencing at least one side-effect and total number of side-effects reported at each time point during the very-low-energy diet phase. Participants with side-effect data at each time point were: day 3–4, *n* = 111; week-1, *n* = 129; week-2, *n* = 129; week-3, *n* = 122; week-4, *n* = 130. Note: some participants reported more than one side-effect at certain time points.

**Table 1 nutrients-16-03125-t001:** Anthropometrics of Fast Track to Health participants who completed the 4-week very-low-energy diet phase.

	Baseline, *n* = 134	4-Weeks, *n* = 134
Weight (kg)	100.0 ± 16.5	94.6 ± 15.9 *
Height (cm)	168.1 ± 9.1	168.2 ± 9.3 ^a^
BMI (kg/m^2^)	35.3 ± 4.1	33.3 ± 4.2 *^,a^
BMI z score	2.4 ± 0.5	2.2 ± 0.4 *^,a^
BMIp95	129.2 ± 15.0	121.5 ± 15.0 *^,a^
Waist circumference (cm)	107.3 ± 11.3	103.3 ± 11.2 *^,b^
Waist-to-height ratio	0.64 ± 0.06	0.61 ± 0.06 *^,b^
Body fat %	43.1 ± 7.6	40.8 ± 7.8 *^,c^

Data expressed as mean ± SD. * indicates significant difference (*p* < 0.05) compared with baseline; ^a^ 1 missing; ^b^ 2 missing; ^c^ 23 missing. Abbreviations: BMI, body mass index; BMIp95, body mass index expressed as a percentage of the 95th percentile; cm, centimeter; kg, kilograms; m, meter; *n*, number.

**Table 2 nutrients-16-03125-t002:** Side effects experienced by participants during the very-low-energy diet phase.

Side Effect, *n* (%)	Day 3–4	Week 1	Week 2	Week 3	Week 4	Total
	*n* = 111	*n* = 129	*n* = 129	*n* = 122	*n* = 130	
Hunger	24 (22)	52 (40)	19 (15)	21 (17)	17 (13)	133
Fatigue	14 (13)	34 (26)	18 (14)	21 (17)	20 (15)	107
Headache	13 (12)	27 (21)	17 (13)	14 (11)	13 (10)	84
Irritability	5 (5)	16 (12)	10 (8)	12 (10)	8 (6)	51
Loose stools	4 (4)	22 (17)	6 (5)	11 (9)	4 (3)	47
Constipation	2 (2)	10 (8)	9 (7)	8 (7)	12 (9)	41
Nausea	2 (2)	7 (5)	6 (5)	2 (2)	6 (5)	23
Lack of concentration	1 (1)	5 (4)	5 (4)	2 (2)	3 (2)	16
Menstrual disturbances, *n* = 67	0 (0)	0 (0)	1 (1)	4 (6)	3 (4)	8
Sensitivity to cold	1 (1)	3 (2)	0 (0)	1 (1)	1 (1)	6
Bad breath	0 (0)	2 (2)	1 (1)	2 (2)	0 (0)	5
Muscle cramps	0 (0)	1 (1)	1 (1)	0 (0)	1 (1)	3
Hair loss	0 (0)	1 (1)	1 (1)	0 (0)	0 (0)	2
Other	10 (9)	22 (17)	11 (9)	9 (7)	6 (5)	58
TOTAL	76	202	105	107	94	584

**Table 3 nutrients-16-03125-t003:** Parent-reported adolescent acceptability of the VLED intervention.

	All of the Time	Most of the Time	Some of the Time	A Little of the Time	None of the Time
They had enough food on the meal plan	33 (30)	53 (47)	22 (20)	4 (4)	0 (0)
They kept to the food prescribed	63 (56)	37 (33)	8 (7)	4 (4)	0 (0)
They ate extra food not prescribed	0 (0)	3 (3)	7 (6)	40 (36)	62 (55)
They didn’t follow the meal plan	8 (7)	11 (10)	3 (3)	18 (16)	72 (64)
They were happy or content during this period	22 (20)	59 (53)	22 (20)	7 (6)	2 (2)
They were frustrated or angry during this period	1 (1)	3 (3)	32 (29)	43 (38)	33 (30)
They felt good about the meal plan ^a^	18 (16)	67 (60)	17 (15)	7 (6)	2 (2)
They felt more in control of their eating habits	32 (29)	58 (52)	18 (16)	4 (4)	0 (0)
They ate differently to the family ^b^	27 (25)	34 (31)	30 (27)	13 (12)	6 (6)
Food habits of some family members changed	7 (6)	35 (31)	40 (36)	16 (14)	14 (13)
Some family members lost weight too ^a^	8 (7)	25 (23)	27 (24)	26 (23)	25 (23)
Preparation of family meals has been easier	13 (12)	32 (29)	40 (36)	13 (12)	14 (13)
Shopping has been difficult on the program	2 (2)	2 (2)	23 (21)	32 (29)	53 (47)
Family members have been supportive of the child in the study	85 (76)	23 (21)	4 (4)	0 (0)	0 (0)
Family life has been difficult for the child in the study	0 (0)	2 (2)	14 (13)	38 (34)	58 (52)
Friends have been interested in the meal plan ^c^	15 (14)	20 (18)	29 (27)	20 (18)	25 (23)
Friends were not supportive of the meal plan ^b^	4 (4)	4 (4)	9 (8)	8 (7)	85 (77)
There have been problems going out with family/friends ^a^	2 (2)	8 (7)	21 (19)	32 (29)	48 (43)
The meal plan has been difficult to follow at school	1 (1)	4 (4)	18 (16)	29 (26)	60 (54)
The study has been disruptive for my child	0 (0)	2 (2)	6 (5)	32 (29)	72 (64)
The study has been beneficial for my child ^a^	82 (74)	22 (20)	4 (4)	0 (0)	3 (3)

Data displayed as *n* (%) for *n* = 112, unless otherwise indicated; ^a^ 1 missing; ^b^ 2 missing; ^c^ 3 missing.

## Data Availability

The raw data supporting the conclusions of this article will be made available by the authors on request due to ethical reasons.

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
