# Peer review of "Efficacy, Safety and Acceptability of a Very-Low-Energy Diet in Adolescents with Obesity: A Fast Track to Health Sub-Study"

_nutrients, 2024, doi:10.3390/nu16183125_

Round 1

Reviewer 1 Report

Comments and Suggestions for Authors

The security and acceptability issues of the treatments for adolescents with obesity are of great interest. The study presents results of a 4-week very low energy diet program, as part of a 52-week randomized controlled trial (Fast Track to Health). The background and introduction are clear and well written. However, I have a number of questions:

Methods

According to the trial Register, recruitment started in 2018 and finished in 2022, and some COVID-19 restrictions occurred during the trial, as mentioned in the manuscript (line 287). Could you indicate how many adolescents were affected by confinements or other restrictions during the 4-week period ?. It will be also useful to know the complete setting, including month and year of study, period of holidays or during school academic period, etc.

I do not see in the reference number 23 the explanation of the “left hand under technique”

I miss details concerning physical activity. Please give information.

Results

Figure 2 is not clear to me. The group of bars on the right “Total number of side-effects” assumes that some participants reported more than 1 side-effect, thus the values at weeks 2 and 3 indicate approximately 1 side effect per subject, while the value at week 1 is between 1 and 2 side-effects. Please confirm.

Discussion

Lines 292-297. It seems that adolescents preferred meetings with the dietitian than receiving text messages. Were the visits with the dietitian in groups?, or the adolescent alone? Or with her/his parents?. I suggest including a comment in the discussion concerning the contact method and the results. In the future, do you advice face to face contact or by telephone/video?, alone, in group?

References. Minor points.

Format of references numbers 21, 29 and 30 should be revised.

Reviewer 2 Report

Comments and Suggestions for Authors

The manuscript is interesting and of value, however, I have some concerns.

Abstract: Please clearly state the aim of the study.

Methods: Please clearly describe the sample you examined, how many individuals took part in the experiment, what was their gender. 

Please include the limitations of your research. 

Reviewer 3 Report

Comments and Suggestions for Authors

The study is done very nicely, with relevant references, and the manuscript reads smoothly.

A couple of relatively minor points may be relevant to making the results more impactful.

Were the assumptions underlying the reported parametric test statistics checked—normality, constant variance, linearity? Reporting evidence that these assumptions are at least approximately satisfied would strengthen the validity of the reported interpretations.

In lines 219-222, the manuscript nots “At 219 all time points, meeting the threshold for ketosis was associated with significantly more weight loss at week-4 (week-1: t=2.645, p=0.010; week-2: t=4.323, p<0.001) week-3: t=2.324, p=0.24; week-4: t=5.817, p<0.001).” Something is not correct with these results, as the reported week 3 t-value is substantially above 2 but the p-value is consistent with a non-significant outcome.
